# Antifungal Peptide P852 Controls Fusarium Wilt in Faba Bean (*Vicia*
*faba* L.) by Promoting Antioxidant Defense and Isoquinoline Alkaloid, Betaine, and Arginine Biosyntheses

**DOI:** 10.3390/antiox11091767

**Published:** 2022-09-07

**Authors:** Chaowen Zhang, Xuan Ou, Jingyi Wang, Zhaoling Wang, Wenting Du, Jianjun Zhao, Yuzhu Han

**Affiliations:** 1College of Animal Science and Technology, Southwest University, Chongqing 402460, China; 2Chongqing Key Laboratory of Herbivore Science, Chongqing 402460, China; 3Chongqing Beef Engineering Technology Research Center, Chongqing 402460, China; 4Chongqing Engineering Research Center of Floriculture, Chongqing 400715, China

**Keywords:** antifungal peptide, green pesticide, antioxidant, metabolome, Fusarium wilt

## Abstract

Green pesticides are highly desirable, as they are environmentally friendly and efficient. In this study, the antifungal peptide P852 was employed to suppress Fusarium wilt in the Faba bean. The disease index and a range of physiological and metabolomic analyses were performed to explore the interactions between P852 and the fungal disease. The incidence and disease index of Fusarium wilt were substantially decreased in diseased Faba beans that were treated with two different concentrations of P852 in both the climate chamber and field trial. For the first time, P852 exhibited potent antifungal effects on *Fusarium* in an open field condition. To explore the mechanisms that underlie P852′s antifungal effects, P852 treatment was found to significantly enhance antioxidant enzyme capacities including guaiacol peroxidase (POD), superoxide dismutase (SOD), catalase (CAT), and the activities of antifungal enzymes including chitinase and β-1,3-glucanase, as well as plant dry and fresh weights, and chlorophyll content compared to the control group (*p* ≤ 0.05). Metabolomics analysis of the diseased Faba bean treated with P852 showed changes in the TCA cycle, biological pathways, and many primary and secondary metabolites. The Faba bean treated with a low concentration of P852 (1 μg/mL, IC_50_) led to upregulated arginine and isoquinoline alkaloid biosynthesis, whereas those treated with a high concentration of P852 (10 μg/mL, MFC) exhibited enhanced betaine and arginine accumulation. Taken together, these findings suggest that P852 induces plant tolerance under *Fusarium* attack by enhancing the activities of antioxidant and antifungal enzymes, and restoring plant growth and development.

## 1. Introduction

The Faba bean (*Vicia faba* L.) is an annual or perennial legume crop grown worldwide for human and animal consumption [1]. A total of about 2.5 million hectares were dedicated to growing Faba bean worldwide in 2018 (FAOSTAT). It generally prefers to grow in areas with moderate climates, such as China, the Middle East, and some Mediterranean countries, where it is recognized as one of the essential food sources [2]. According to both the annual output and total planted area, China is the world’s largest producer of Faba bean, accounting for 34.5% of the global production [3]. Faba bean is nutritious because of its high content of proteins and vitamins, which imparts its versatile food, feed, and fodder applications [4]. Moreover, the Faba bean is a model plant for environmental pollution, which can be used to detect and evaluate environmental pollution and toxicity. The chromosome haplotype of the Faba bean is six pairs of fairly large chromosomes, which are well suited for microscopic observation, especially at its roots [5].

*Fusarium oxysporum* can infect a wide range of commercial crops and can live in soil for years without a host, making it difficult to eradicate [6]. Researchers around the world have tried to combat Fusarium wilt by employing a wide range of approaches, including the development of *Fusarium* resistant crops, the use of chemical fungicides, improving tillage methods, and biological control strategies [7,8]. It is desirable to employ *Fusarium* resistant varieties, but it will take a substantial financial and time commitment to develop new varieties with built-in resistance. Chemical pesticides are the most widely used method, despite the fact that they pose threats to human and environmental health when used in excess. The use of 10 g/L carbendazim, for instance, had a significant inhibitory effect on cashew Fusarium wilt incidence; however, its overuse was found to be harmful to the environment and food safety [9]. Although biological control can safeguard the environment, it is generally less effective. Among the sporadic reported successes, symbiotic microbial combination agents enhanced the activities of numerous antioxidant enzymes and effectively reduced the incidence of the tomato *Fusarium* root rot disease [10].

In the Faba bean, Fusarium wilt is one of the main devastating diseases that pose threats to high-yield and broad acreage cultivation, which is primarily caused by a soil-borne fungal pathogen *F. oxysporum* f. sp. *fabae* (FOF) [11]. Faba bean varieties grown in Southwest China are largely susceptible to *F. oxysporum*, and an effective strategy to control the disease is clearly lacking. Symptoms of Faba bean Fusarium wilt caused by *F. oxysporum* include the yellowing and wilting of the leaves, which eventually turn black and die. The vascular bundle system of the root system and stems turns brown to black, with discoloration and decay at the base of the root system and stems [1,6]. It was reported that intercropping with wheat significantly reduced the incidence of Faba bean Fusarium wilt while increasing the activities of guaiacol peroxidase (POD), catalase (CAT), chitinase, and β-1,3-glucanase (β-1,3-GA) enzymes [11,12]. Then, it was discovered that wheat extracts were able to reduce the incidence of Faba bean Fusarium wilt and enhance the activity of antioxidant enzymes in the Faba bean [11]. Further, silver nanoparticles (AgNPs) treated with the aqueous leaf extract of *Melia azedarach* have a good control effect on the tomato Fusarium wilt [13]. These findings led to the pursuits of a “green pesticide” to control Faba bean Fusarium wilt, which has low toxicity and minimal environmental concerns [14]. While it remains hitherto unknown about the causative components against Fusarium in these studies, it was found that antimicrobial peptides (AMPS) rich in lysine (Lys) or arginine (Arg) and tryptophan (Trp) found in numerous cellular organisms are antimicrobial to various pathogens, including *F. oxysporum* [15]. The anti-*Fusarium* components in *Bacillus amyloliquefaciens* were also identified as AMPS [16]. At present, AMPS secreted by *Bacillus* spp. are the main antimicrobial components that effectively inhibit pathogens [17,18]. We have previously reported the isolation of a cyclic peptide termed P852 because of its molecular weight of 852.44 Da in the fermentation broth of *Bacillus velezensis* strain L-H15 [19]. P852 can antagonize a bewildering array of pathogenic fungi, causing cell membrane and organelle damage and hence strongly inhibiting the growth and reproduction of *F. oxysporum*. It appears that its fungal inhibitory activity is more potent than that of the fungal antibiotic, amphotericin B [19].

Although P852′s high efficacy against Fusarium wilt caused by *F. oxysporum* suggests that it could be considered a green pesticide, all the data are so far collected from the laboratory conditions, and it remains an open question whether it would be as effective in the open field. In this study, we first appraise the comparative effects of P852 against the Fusarium wilt of Faba bean in experimental fields and laboratory settings. Then, we systematically conducted and noted the plant growth and physiological experiments and analyzed a range of parameters including plant height, root length, dry weight, fresh weight, and antioxidant enzyme activity in the Faba bean infected by *Fusarium* upon treatment with the P852 peptide. We aim to explore the mechanisms that underlie P852′s effects in reducing the incidence and biomass of Faba bean Fusarium wilt by using nontargeted metabolomics with liquid chromatography with tandem mass spectrometry (LC-MS/MS) to assess plant growth and physiological alterations in Faba bean Fusarium wilt treated with the P852 peptide.

## 2. Materials and Methods

### 2.1. Experimental Materials and Experimental Design

Seeds of Faba bean ‘Tong chanxian’ were purchased from the Chongqing Hechuan District Loofah Vegetable Research Institute, Chongqing, China. *F. oxysporum* was isolated from *V. faba*, and its mycelia were stored in the Forage Microbe Laboratory, Southwest University, Chongqing, China, at 4 °C, until use.

P852 antifungal peptide was extracted in the fermentation broth of the *Bacillus velezensis* strain L-H15 (GenBank accession number CP010556.1) [17]. The optimized P852 antifungal peptide’s conditions and extraction methods were as per the previously described [18]. The primary structure of the P852 antifungal peptide was deduced to be glutamine (Gln)-isoleucine or leucine (Ile or Leu)-glycine (Gly)-glycine (Gly)-isoleucine or leucine (Ile or Leu)-isoleucine or leucine (Ile or Leu)-threonine (Thr)-valine (Val) sequence by thin-layer chromatography (TLC), matrix-assisted laser desorption time-of-flight mass spectrometry (MALDI-TOF-MS), electrospray-triple quadrupole mass spectrometry-time-of-flight tandem mass spectrometry (ESI-Q-TOF-MS), and nuclear magnetic resonance (NMR) [19].

Location of the experiment and inoculation method: The experiment was conducted in the Forage Microbe Laboratory, College of Animal Science and Technology, Southwest University. The plants of the Faba bean were cultivated in an artificial climate chamber (Phase 1: at 25 ℃, 70% relative humidity (RH), 14,000 lx (light intensity); Phase 2: 75% RH, 0 lx, at 16 ℃) using the Hoagland Nutrient Solution (pH = 6.5) with sterile quartz sands in a 500 mL tissue culture bottle until the plants grew 7–8 true leaves. Then, the plant roots (total n = 36) were excised and inoculated with FOF at 1 × 10^6^ CFU/mL spores suspension following the root-excision inoculation method [20], and the P852 solution (sterile water) was poured into the root bases of the Faba bean after two weeks.

Experimental design: The experiment consisted of one control (CK, treated with sterile water) and two treatments containing IC_50_ of P852 (1 μg/mL) and MFC (10 μg/mL, H) of P852. As described by Han [19], 10 g/L carbendazim was used as a positive control to treat Fusarium wilt in Faba bean leaves and roots [9].

### 2.2. Field Trials

In February 2022, a field trial was conducted in Rongchang Forage Germplasm Fields, Southwest University, Chongqing, China (105°17′–105°44′ E, 29°150′–29°41′ N). In this area, the total annual accumulated temperature is 6883 °C, the average altitude is 308 m, the annual average precipitation is 1118.3 mm, the annual photoperiod is 1083 h, and the RH is 72%. The annual average temperature is 17.8 °C, and the frost-free period is 327 days (d); the soil in the experimental site is gray–brown–purple soil, and the soil texture is medium loam, with a soil pH of 6.5 [21]. The organic matter content is less than 0.5%, ammonium nitrogen is 16.0 mg/kg, available phosphorus is 12.5 mg/kg, and available potassium content is 85.6 mg/kg in the field. In this field, disease incidence occurred in the Faba bean that were 90 days old, with the natural occurrence of Fusarium wilt. The experiment was conducted in a randomized block design, and repeated three times, with an area of 1 m^2^ in each plot (Figure 1). No insecticides, fungicides, or herbicides were used during the growing period. The other management was based on local agronomic practices.

### 2.3. Evaluation of the Incidence of Fusarium Wilt

Faba bean were inoculated with FOF, and maintained in an artificial climate chamber for 35 days. In the field, disease incidence occurred in Faba beans that were 90 days. The disease grade assessment in the Faba bean was performed as previously described [1].The survey of the disease incidence and disease index of Faba bean’ s Fusarium wilt was conducted, and 16 plants were investigated in each treatment of the artificial climate chambers, 3 of fields were randomly selected for each same treatment group, and 12 plants were taken from each field using the five-point sampling method. The degree of disease incidence was divided into 5 levels: level 0—no symptoms; level 1—localized slightly diseased spots or slightly discolored spots on the stem base or roots; level 2—diseased spots on stem base or main lateral roots, but not contiguous; level 3—diseased spots, discoloration or rot on 1/3~1/2 of stem base or roots; level 4—stem base surrounded by diseased spots or most of the root system discolored and rotted; level 5—plants died. Incidence and disease index were calculated.
$$\mathrm{Incidence}=\frac{\mathrm{Number}\;\mathrm{of}\;\mathrm{diseased}\;\mathrm{plants}}{\mathrm{total}\;\mathrm{number}\;\mathrm{of}\;\mathrm{plants}\;\mathrm{investigated}}\times 100\%\;\mathrm{Disease}\;\mathrm{index}=\frac{\sum \left(\mathrm{Number}\;\mathrm{of}\;\mathrm{diseased}\;\mathrm{plants}\;\mathrm{at}\;\mathrm{each}\times \mathrm{level}\right)\;}{\mathrm{The}\;\mathrm{highest}\;\mathrm{level}\;\times \mathrm{total}\;\mathrm{number}\;\mathrm{of}\;\mathrm{plants}\;\mathrm{investigated}\;}\times 100$$
$$\mathrm{Disease}\;\mathrm{index}=\frac{\sum \left(\mathrm{Number}\;\mathrm{of}\;\mathrm{diseased}\;\mathrm{plants}\;\mathrm{at}\;\mathrm{each}\times \mathrm{level}\right)\;}{\mathrm{The}\;\mathrm{highest}\;\mathrm{level}\;\times \mathrm{total}\;\mathrm{number}\;\mathrm{of}\;\mathrm{plants}\;\mathrm{investigated}\;}\times 100$$

Note: Number of diseased plants: Survey finds number of the Faba beans with Fusarium wilt; Level: Level refers to the number of incidence levels of the Faba bean Fusarium wilt.

### 2.4. Measurement of Chlorophyll Content

Chlorophyll content of the Faba bean was measured at 7 d after the initiation of treatment with P852 peptide, as previously described [22,23]. Briefly, leaf samples were collected and washed with running tap water prior to being cut into thin strips with scissors. Then, 25 mL 95% alcohol was added to 0.2 g of leaf samples in a tube, which was vigorously shaken, and incubated in the dark for 24 h. A 95% alcohol solution without leaf samples was used as the control group (CK). Chlorophyll extracts thus obtained were measured on a spectrophotometer (UV Spectrophotometer, Shanghai Jinghua Technology Co., Ltd., Shanghai, China). The absorbance values were read at wave length of 663, 646, and 470 nm. Chlorophyll extracts at 663 and 645 nm could be depicted as follows: Ca(Chlorophyll a) = 13.95 × A_665_ − 6.88 × A_649_ and Cb(Chlorophyll b) = 24.96 × A_649_ − 7.32 × A_665_, Carotenoid = 1000 × A470 − 2.05 × Ca − 114.8 × Cb)/245, and the chlorophyll content were expressed as mg/g.

### 2.5. Measurement of Plant Physiological Parameters

Seven days after the initiation of the treatment with P852 peptide, the plant height and root length were measured using a Vernier caliper. The fresh weight of the aboveground biomass and roots was determined by an electronic balance. To determine the dry weight, the samples were treated in an oven at 105 ℃ for 20 min before being dried at 75 ℃ for 24 h [24].

### 2.6. Measurement of Antioxidant Enzyme Activities

Faba bean leaves and roots in climate chambers were collected at 24 h, 48 h, 72 h, and 96 h after treatment with P852 peptide. The antioxidant enzyme activities of POD and superoxide dismutase (SOD), chitinase, and β-1,3-GA were assayed when Faba bean leaf and root samples were treated with P852 peptide for 24 h, using the spectrophotometer as previously described, with some modifications [1,22,25].

To determine POD activity, an appropriate amount of phosphate buffer was added to 0.2 g of the plant leaf or root samples (per sample *n* = 3, from different plants) and ground to a powder with mortars and pestles. The grinding fluid was then loaded into a centrifuge tube and centrifuged at 3000× rpm for 10 min. The supernatant was transferred into a 25 mL volumetric flask. In a tube, 0.1 mL of the enzyme was mixed with 2.9 mL of 0.05 M phosphate buffer (pH = 5.5), 1 mL of 2% H_2_O_2_, and 1 mL of 0.05 M methylcatechol, and the enzyme solution was boiled for 5 min and used as a control (CK). Then, the tubes were incubated in a thermostated water bath at 34 ℃ for 3 min. The absorbance value was measured using a spectrophotometer every five minutes, with a POD activity unit (U/g·min) at a 0.01 ΔA_470_ value change per minute [1].

The nitro-blue tetrazolium (NBT) method was used to determine SOD activity [1]. About 0.2 g of the fresh plant leaf and root tissues (*n* = 3) were left at 4 ℃ before being ground to a pulp in mortars with phosphate buffer and centrifuged at 4000 rpm for 10 min. Then, the extract was added to the enzyme solution containing 0.05 mL enzyme (control was distilled water), 0.3 mL of 20 μM riboflavin solution, 0.3 mL of 100 μM EDTA-NA_2_ solution, 0.3 mL of 750 μM NBT solution, 0.3 mL of 130 mM methionine solution, 1.5 mL of 0.05 M phosphate buffer (pH 7.8), and 0.25 mL of distilled water in a 5-mL finger tube. Following a thorough mix, the control group was incubated in the dark, and the remaining tubes were incubated under 4000 l× light for 20 min. The absorbance was measured using a spectrophotometer at 560 nm, with 50% inhibition of NBT photoreduction as an activity unit of SOD enzyme (U/g).

Chitinase activity was determined using the chitinase Assay kit (Solarbio Science & Technology Co., Ltd., Beijing, China), following the instructions from the manufacturer. The amount of enzyme that decomposes chitin to produce 1 μmol *N*-acetyl-D-(+)-glucosamin per gram of tissue per hour is one enzyme activity (U/g) unit at 37 °C. β-1,3-GA activity was measured using the β-1,3-GA Assay kit from (Solarbio) following the manufacturer’s instructions. One microgram of the reduced sugar produced per gram of tissue per hour was defined as one unit of enzymatic activity (U/g).

Catalase (CAT) activity was measured by the KMnO_4_ titration method, as previously described [22]. A total of 0.2 g of the fresh plant leaf and root tissues (*n* = 3) were collected at 24 h, 48 h, 72 h, and 96 h and ground to homogenate in mortars with phosphate buffer (pH = 7.8). The homogenate was centrifuged at 4000× rpm for 15 min. Then, 2.5 mL of the enzyme supernatant was added to 2.5 mL of 0.1 M H_2_O_2_ and 10% H_2_SO_4_ and placed in a water bath at 30 °C for 10 min. The mixed solution was titrated with 0.1 M KMnO_4_ until it appeared pink. CAT activity (mg/g·min) is expressed in milligrams of H_2_O_2_ decomposed in one minute per gram of fresh weight sample.

### 2.7. Malondialdehyde (MDA) Content and Electrolyte Leakage

Faba bean leaves and roots in climate chambers were collected at 24 h, 48 h, 72 h, and 96 h after treatment with P852 peptide. The malondialdehyde (MDA) content was determined by the thiobarbituric acid (TBA) reaction, as previously described [25]. A total of 0.5 g of fresh leaves and roots (*n* = 3) were homogenized with 5% trichloroacetic acid (TCA) and centrifuged at 3000 rpm for 10 min and 2 mL of the supernatant was boiled with 2 mL 0.67% TCA for 30 min. After centrifugation at 4 ℃, the supernatants were taken for measuring absorbance with a spectrophotometer at 450 nm, 532 nm, and 600 nm.

Electrolyte leakage (EL) was estimated by a conductivity meter and a DDS-11A conductivity meter (Shanghai Leici, Shanghai, China). A total of 0.5 g of leaf or root samples (*n* = 3) were cut in tubes with 20 mL distilled water for 3 h at 32 °C. Then, electrical conductivity (EC1) was measured. After placing the tubes in a boiling water bath (100 °C) for 20 min, the tubes were allowed to cool, and the final EC2 was determined [26]. EL was calculated using the formula EL (%) = EC1/EC2.

### 2.8. Metabolite Extraction and Metabolite Profiling

Faba bean leaf and root samples were collected from the plants in climate chambers 96 h after treatment with the P852 peptide. The metabolites were extracted as previously described with slight modification [27,28]. Briefly, the fresh plant samples were freeze-dried under a vacuum by the LGJ-10 lyophilizer for 24 h (Songyuan Freeze Dryer, Shanghai, China) and powdered with a 50 Hz grinder. Then, 50 mg of the powdered samples thus obtained were added to 0.6 mL of 70% methanol, decanted into 2 mL EP tubes, and placed at 4 °C overnight in a refrigerator. This was followed by sonication for 5 min prior to centrifugation at 12,000× *g* for 10 min. The supernatant was filtrated using a Biosharp filter BS-QT-013, and 0.22 μm pore size (Biosharp Life Sciences, Hefei, China). From each sample, 10 μL was taken and pooled as the quality control (QC) [29]. We will insert one QC sample in every five samples for monitoring method stability and data reliability (Figure 1).

LC–MS/MS analyses were implemented using an ultra-high performance liquid chromatography (UHPLC) UltiMate^®^ 3000 (Dionex, Sunnyvale, CA, USA) fitted with a UPLC Hypersil GOLD C18 column (2.1 × 100 mm, particle size 1.9 μm) (Thermo Fisher Scientific, Waltham, MA, USA) that was coupled with a Q-Exactive Orbitrap (Thermo Fisher Scientific). The flow rate was 0.2 mL/min, and the column temperature was 35 °C. The injection volume was 2 μL. The mobile phase containing solvent A (ultrapure water with 0.1% formic acid) and solvent B (0.1% formic acid in methanol), solvent C (ultrapure water with 0.1% NH_3_), and solvent D (0.1% NH_3_ in methanol) was eluted in positive ion mode with the following gradient program: 0–10 min with 5% B and 95% A; 10–12 min with 5% A and 95% B; 12–13 min with 5% A and 95% B; and 13.1–14 min 95% A and 5% B, eluted in a negative ion mode with the following gradient program: 0–2.5 min with 95% C and 5% D, 2.5–16.5 min with 95% D and 5% C, 16.5–19 min with 95% D and 5% C 19, and 20 min with 95% C and 5% D.

The Q-Exactive Orbitrap was used to acquire MS/MS spectra in an information-dependent acquisition (IDA) mode under the control of the acquisition software Xcalibur (Thermo Fisher Scientific, Waltham, MA, USA). The HESI source operation parameters were as follows: sheath and aux gas flow rates of 40 and 10 Arb [30], respectively, capillary temperature of 320 °C, and a full mass scan (*m*/*z* 70–1050) with a resolution of 70,000. The MS/MS scanning mode was set as a data-dependent ms2 (dd-ms2) scan at a resolution of 35,000. The high collision dissociation was set to 20/40/60 eV in the NCE mode [31]. The spray voltage was 3.5 kV (positive ion mode)/−2.5 kV (negative ion mode).

### 2.9. Statistical Analysis

The data in the table are the mean ± SE. IBM SPSS statistics 26.0 (SPSS, Chicago, IL, USA) was used for the one-way analysis of variance (ANOVA) and Duncan’s *t*-test when *p* < 0.05 indicated a statistically significant difference. GraphPad Prism software version 8.0 (GraphPad Software Inc., San Diego, CA, USA) was used. Row data from untargeted metabolomics were imported into compound discover 2.1 (Thermo Fisher Scientific, Waltham, MA, USA). Peak detection, extraction, deconvolution, normalization, peak alignment, and metabolites were matched by the Mz Cloud and mzVault databases. The metabolite data set was generated, including the mass-to-charge ratio, retention time, and peak area. The positive and negative data were merged to obtain data that were imported into SIMCA-P 14.1 software (Umetrics, Umea, Sweden) for multivariate statistical analysis, including principal component analysis (PCA) and orthogonal partial least-squares-discriminant analysis (OPLS-DA). Potential differential metabolites were screened according to variable projection importance in projection (VIP) ≥ 1, *p* < 0.05 and fold change (FC) > 1.5. Differential metabolic chemicals were analyzed using Metaboanalyst 5.0 (https://www.metaboanalyst.ca/) (accessed on 5 May 2022) to obtain vital metabolic pathways. Moreover, the KEGG database (https://www.kegg.jp/kegg/pathway.html) (accessed on 21 June 2022) was utilized to annotate and construct the pathway.

## 3. Results and Discussion

### 3.1. Evaluation of the Incidence of Fusarium Wilt

Under the artificial climate chamber condition, the incidence of Fusarium wilt in the Faba bean was significantly reduced 0.36 times, 0.36 times, and 0.27 times, while its disease index was significantly reduced 0.48 times and 0.64 times, and 0.43 times in the D (1 μg/mL), H (10 μg/mL) of antifungal peptide and 10 g/L of carbendazim treatment compared to the CK group, as shown in Table 1 (*p* ≤ 0.05). Likewise, under the field experiment, the incidence of Fusarium wilt was significantly reduced by 0.36 times, 0.33 times, and 0.30 times, respectively, and its disease index decreased by 0.51 times, 0.57 times, and 0.41 times in the D, H, antifungal peptide, and 10 g/L carbendazim treatments compared to the CK group, as shown in Table 1 (*p* ≤ 0.05). The disease incidence and index of Faba bean Fusarium wilt were most significantly reduced in Faba bean under the H and D antifungal peptide treatments (Table 1), which are well in line with our previous finding that a robust inhibitory effect of peptide P852 against numerous plant fungal pathogens [19]. Since the control effect exceeds that of carbendazim, it is unlikely to cause environmental pollution and toxicity. This is also congruent with a previous studies that the recombinant peptides extracted from tobacco plants exhibited a significant inhibitory effect against *F. oxysporum* [32], and the cyclic lipopeptides isolated from *Paeni Bacillus polymyxa* were inhibitory to *Fusarium moniliforme* [33]. However, both reports were based on laboratory-derived data. In fact, to our knowledge, the present findings represent the first field assessment of AMPS against filamentous fungi.

### 3.2. Effect of Different Treatments on Faba Bean Plant Growth and Chlorophyll Content

D and H treatments bolstered Faba bean growth relative to CK, as indicated by the increases in chlorophyll content, plant height, dry weights, both dry and fresh weights of above ground plant biomass, both dry and fresh weights of roots, and stem diameter (Table 2). This antifungal peptide extracted from *B. velezensis* has been considered as the primary antifungal substance in *B. velezensis* [34]. The present investigation determined the specific concentrations of the antifungal peptide that are effective in inhibiting *Fusarium* and reinvigorating the plant growth, which is manifested by the significant increases in most of the growth indicators in Faba bean treated with both H and D concentrations of the antifungal peptide.

### 3.3. Effect of Antifungal Peptide P852 Treatments on Antioxidant Enzyme Activities, MDA Content, and Electrolyte Leakage

To determine the effect of antifungal peptides on the Faba bean with Fusarium wilt based on the biochemical point, the activities of POD, SOD, CAT, and the contents of MDA and EL in the leaves and roots of the Faba bean treated with different concentrations of the AMPS were determined. In the leaves of the Faba bean that were subjected to 24 h treatments of D and H, the fold change rate of POD activity increased by 0.84 and 0.75 times, respectively, relative to the CK group (Figure 2). Interestingly, the maximum effect was achieved by the treatment for 24 h, whereas there was no significant difference between the D and CK at 96 h (*p* ≤ 0.05). In contrast, the fold change rate of the H treatment’s POD activity increased by 1.7 times relative to CK at 48 h, which is consistent with Pan et al. that the mixed use of pesticides and foliar fertilizer could significantly raise POD activity within 7 days [28]. The SOD activity of the Faba bean subjected to D and H treatments was increased by 0.48 times and 0.37 times, respectively, relative to CK, following treatments for 96 h, which was the maximum effect observed. Interestingly, after 24 h, the SOD enzyme activity in plants treated with 10 g/L carbendazim group was lower than CK. Moreover, significant increases in CAT enzyme activity were observed following the treatment with H relative to other groups.

MDA content and the EL of the plants treated with 10 g/L of the carbendazim group and CK group reached their maximum at 96 h, which is in accordance with the data of SOD enzyme activity. It is likely that carbendazim damages the antioxidant system of plants [35], whereas the treatments with D and H significantly reduced MDA content and EL compared to CK.

SOD, POD enzyme activity, EL, and MDA content were differentially displayed in the leaves and roots of plants (Appendix A). The POD activities were significantly increased by 0.63 times and 0.51 times in the D and H treatments, respectively, compared with the CK group. In contrast, the SOD activities in the H and D treatment groups were not significantly different from those in the CK group. D and H group treatments significantly reduced EL and MDA contents by 0.25, 0.66 and 0.38, 6.09 times, compared with CK (*p* ≤ 0.05).

Generally, when plants are subjected to biotic and abiotic stress factors, a series of stress responses activate the antioxidant system due to the production of reactive oxygen species [36]. Increasing research is focused on exogenous antioxidants, which can effectively alleviate biotic and abiotic stresses and enhance antioxidant enzyme activities in plants [37,38]. The invasion of plants by phytopathogenic fungi causes a massive accumulation of reactive oxygen species (ROS) [39]. Plant antioxidant enzymes such as SOD, POD, and CAT can scavenge ROS in plants, thereby alleviating environmental stresses, which have been commonly used as reliable indicators of plant disease resistance in plants [1]. SOD is a key enzyme that is the first line of defense against ROS [25] and transforms superoxide radicals into less toxic reagents, generating H_2_O_2_ [37,40]. CAT and POD are essential enzymes that detoxify excess ROS by catalyzing the decomposition of H_2_O_2_ into water and converting divalent oxygen [41]. MDA is the end-product of the oxidation of polyunsaturated fatty acids, directly displaying the extent of lipid damage caused by oxidative stress [42]. EL increases under plant stress, leading to impaired cell membrane integrity [43].

### 3.4. Effect of Antifungal Peptide P852 Treatments on β-1,3-GA and Chitinase Activities

The activities of β-1,3-GA and chitinase were measured after the P852 treatments (Figure 3A,B). Compared with the CK group, β-1,3-GA activities treated with D and H treatments in the leaves of Faba bean with Fusarium wilt were significantly enhanced by 233% and 92%, and the roots were significantly increased by 212% and 127%, respectively. The chitinase activities of leaves significantly increased by 27% and 67%, and the roots were significantly enhanced by 51% and 57% at D and H treatments in the leaves, respectively, compared with the CK group, as shown in Figure 3 (*p* ≤ 0.05).

When pathogenic fungi infect the host crop, the self-defense response is initiated, and with the expression of disease-related proteins, plants can resist the invasion of plant pathogens by releasing defense substances such as chitinase and β-1,3-GA, which can degrade the cell walls of pathogenic fungi [1]. Chitinase, the main component of the fungal cell wall, prevented mycelial growth, causing rough deformities, and even complete cell lysis, thereby mitigating fungal infestation [44]. β-1,3-GA, which is found mainly in plants, catalyzes the hydrolysis of β-1,3-glucan bonds and induces the synthesis of large amounts of β-1,3-GA in cells when plants are under fungal attack, inhibiting the growth of mycelium, damaging the fungal cell wall and ultimately leading to its death [45]. The present findings corroborate previous studies that methyl jasmonate can significantly improve chitinase and β-1,3-GA activity [46]. To date, the effects of *Bacillus* spp. AMPS on chitinase and β-1,3-GA activities have rarely been reported.

### 3.5. Effect of P852 Treatments on the Overall Metabolic Profile of Faba Bean Leaf

To better comprehend the mechanism that underpins Fusarium wilt suppression by P852 in Faba bean, the metabolomics of the leaf tissues derived from plants subjected to D and H P852 treatments were analyzed using UHPLC-Q-Orbitrap-MS. A total of 967 different metabolites were authenticated via untargeted metabolomic analysis. The normalized data were sorted and converted into CSV format for import into the SIMCA software. Based on unsupervised PCA results (Figure 4A), the PCA score plot for leaves indicated that the H and D groups were separated distinctly from the CK group, which was in line with the Faba bean growth biomass and antioxidant enzyme activities. The first principal components (PC1) and second principal components (PC2) explained 42.1% and 24.3% of the total variance in all samples, respectively. Further, the *p* value of Duncan’s *t* test (*p* < 0.05) and VIP of OPLS-DA plots (VIP > 1) function as standards for screening metabolic differences. To visualize the expression patterns in response to the changes of P852, Venn diagrams were generated to find the associated changes in metabolites under different treatments (Figure 4B). Sixty-eight common metabolites were expressed with different treatments (H vs. CK, D vs. CK, H vs. D), and 92 common metabolites were generated in response to D and H P852 treatments.

Orthogonal projection to latent structures-discriminate analysis (OPLS-DA) is a multivariate statistical analysis method with a supervised pattern recognition that can effectively eliminate the effects unrelated to the study and thus filter out differential metabolites [47]. OPLS-DA score plots were performed by SIMCA software, including the groups of H, D, and CK, as shown in Figure 5A–C, which denoted separation between groups. The permutation test was utilized to evaluate and validate the quality of the OPLS-DA model [28]. Values is close to 1 for R2 and more significant than 0.5 for Q2 implied that the OPLS-DA model was stable and had a good ability for fitness and prediction [48]. Then, permutation plots demonstrated that this model had satisfactory discriminatory differences between H vs. CK, D vs. CK and H vs. D (Appendix A) after 200 permutations [49]. However, our results had minor overlap, and D vs. CK of Q2 was slightly less than 0.5, and the Q2 intercept values of all permutation plots were less than zero (Appendix A). This demonstrated that the model fits well [49]. It is therefore that the OPLS-DA models are compatible with identifying the differences between groups.

Volcano plots were utilized to visualize the *p* value of Duncan’s *t*-test, the fold change (FC) value, and the VIP value to filter different metabolites; metabolites were screened by *p* values < 0.05, VIP values > 1, and FC values > 1.5. As shown in Figure 5D–F, the red part represents the upregulation of differential metabolites, the green part represents the downregulation of differential metabolites, and the grey part represents nonsignificant metabolites. H treatment resulted in most upregulated metabolites compared with other treatments, with 93 metabolites upregulated and 44 downregulated. D treatment led to the upregulation of 53 metabolites and downregulation of 53 metabolites. The fact that H treatment upregulated more metabolites than D treatment may suggest that H treatment is more effective in suppressing Fusarium wilt in Faba bean. Previous studies reported that soybean endogenous AMPS could elicit a variety of defense signaling pathways, such as salicylic acid and jasmonic acid pathways, to mediate resistance responses to enhance tolerance to the *Phytophthora* root and stem rot in the soybean [50].

### 3.6. Metabolic Pathways in Leaves of Faba Bean in Response to P852 Treratment

MetaboAnalyst 5.0 was used to create biological metabolic pathways (Figure 6), and values greater than 0.1 were considered to be potential target pathways based on their impact on metabolic pathways [49]. Major enriched metabolic pathways of differential metabolic compounds were analyzed using KEGG databases; these pathways included arginine biosynthesis, isoquinoline alkaloid biosynthesis, phenylalanine (Phe) metabolism, tyrosine biosynthesis, and Trp biosynthesis under D treatment, and arginine biosynthesis, betaine biosynthesis, Phe metabolism, alanine aspartate, and glutamate metabolism under H treatment.

A heatmap was generated to visualize the alterations in differential compounds screened with VIP > 1 (OPLS-DA) and *p* < 0.05 values as different compounds. Heatmaps are a versatile method of depicting both individual data packets and the population level clustering of pattern changes [47]. Positive and negative ESI modes show the presence of a total of 61 different metabolites, including 15 different amino acids and their derivatives, 10 different fatty acids, 9 different organic acids, 5 different secondary metabolites, and 22 other metabolites. As can be observed in Figure 7, 61 differentially identified metabolites were visualized as five heatmaps, with red indicating upregulated metabolites and blue indicating downregulated ones. Interestingly, amino acids respond differently to high and low concentrations of the antifungal peptide P852 (D and H treatments, Figure 7A). This is well in line with the previous studies stating that amino acids play a crucial role in regulating plant growth and development, stress resistance, and easing external biotic and abiotic stresses [51,52,53]. The content of arginine, asparagine, proline (Pro), and pyroglutamic acid was higher than that in the control group under the H treatment, which is consistent with a previous report that exogenous NO can increase the content of arginine and Pro in tomato seedlings under copper stress [54]. D treatment led to marked increase in the levels of tyrosine, 2-hydroxyphenyl, norleucine, citrulline, ornithine, glutamic acid, and pyroglutamic acid. The metabolism of glutamate to ornithine, arginine, and Pro was the leading network of nitrogen metabolism pathways in plants, and it also produced some intermediates, which played a crucial role in plant development and stress [55]. Upon stress, the induced Pro is capable of eliminating ROS and constraining lipid peroxidation, thereby alleviating stress in plants [56,57]. It is conceivable that P852 may induce this class of amino acids and protect against biotic stress in the Faba bean.

Aromatic amino acids (Trp, tyrosine, and Phe) are involved in the shikimate-phenylpropanoid pathway, betaine biosynthesis pathway, and auxin biosynthesis, as shown in Figure 7. Compared with the control group, the content of aromatic amino acids was increased by H treatment, whereas D treatment was relatively reduced, consistent with the study of Pan et al. [28]. Cucumbers treated with pesticides had a higher concentration of aromatic amino acids in their fruit [23], a finding that may have stimulated the shikimate-phenylpropanoid pathway of auxin biosynthesis [58]. Both D and H treatments induced auxin biosynthesis, of which tryptophan is a key component, providing a mechanistic basis for the enhancement of plant growth and biomass accumulation. This finding is congruent with previous studies that demonstrated flavonols affecting auxin synthesis, thereby promoting plant growth and development [59].

Furthermore, the metabolome revealed the alteration in fatty acid composition in Fusarium wilt-infected Faba bean leaves under the H and D treatments of P582. The contents of α-linolenic acid and 16-heptadecene-1,2,4-triol in Faba bean leaves were increased under the H treatment. In contrast, the fatty acid composition in the D treatment group was similar to that in CK (Figure 7B). Alpha-linolenic acid is widely present in green plants in the form of glycerides and is essential for plant growth and adaptation to environmental pressures [60]. H treatment of P852 can induce the biosynthesis of α-linolenic acid that is the precursor of jasmonic acid, leading to Fusarium resistance (Figure 8). Such a premise supports the previous study that treatment with hexanoic acid can induce the α-linolenic acid pathway to generate relevant antibacterial substances, as demonstrated by the C13 labeled hexanoic experiment [61].

In contrast to H treatment, the levels of organic acids, especially oleanolic acid and 2-hydroxycinnamic acid, were barely increased under D treatment, relative to CK (Figure 7C). It was found that numerous plant growth regulators and carbon sources could alter oleanolic acid levels in *Polyscias fruticosa* pericarp tissue suspension culture [62], and it is generally recognized to be present ubiquitously in plants participating in plant defenses and defense systems against pathogens [63,64]. The contents of other organic acid, including benzoic acid, critic acid, caffeic acid, malic acid, and gluconic acid were all elevated under the H treatment. The elevation in the level of benzoic acid, which is a synthetic agent of salicylic acid, is conducive to the biosynthesis of salicylic acid (Figure 8), which plays an essential role in the disease resistance signaling pathway [10,65]. Among the elevated organic acid by the high concentration of P852 treatment, caffeic acid is a common antioxidant in plants [66]. As a product of the TCA cycle, malic acid is thought to play a vital role in plant metabolism as a reducing agent, osmolyte, and pH regulator [67]. It was found that the ratio of malic acid and citric acid was increased in the seedlings’ exposure to NaCl stress, suggesting that more energy generated by the TCA cycle is required to maintain the salinity-induced redox homeostasis [26,68]. Likewise, high concentrations of P852 have reinforced energy generation to preserve redox homeostasis in Fusarium wilt-infected Faba bean leaves.

It is recognized that secondary metabolites can assist plants in adapting to adverse environmental conditions, such as drought and high temperatures. As shown in Figure 7D, D treatment of P852 induced a significant up-regulation of isoquinoline, which is one of the most common alkaloids in the plant kingdom [69]. Isoquinoline compounds can significantly increase antioxidant enzyme activities in plants [70,71,72]. The enhanced accumulation of isoquinoline compounds may be associated with elevation in antioxidant enzyme activities to scavenge ROS and inhibit the infestation of plant pathogens. DL- stachydrine is an alkaloid mainly found in *Leonurus japonicus* Houttuyn, which was induced by arbuscular mycorrhizal fungi colonization in poplar seedlings to defend against insects [73]. The substantial increase in stachydrine as a result of D and H treatments represents the first observation that stachydrine is associated with the defense against a fungal pathogen. Upon P852 treatment, the relative content of betaine was increased, which is a cytoplasm-localized substance that is capable of increasing the activities of antioxidant enzymes and rapidly scavenging the ROS produced upon fungal attack [74,75].

Despite no prior study finding that exogenous AMPS can induce betaine production in plants infected with Fusarium wilt, it is conceivable that H treatment may induce alkaloids’ production as a protective strategy.

In the control group, the relative contents of phytosphingosine, carnitine, and arachidonic acid were significantly increased (Figure 7E). This is consistent with the notion that plants produce these defense-related compounds upon biotic stress [76,77,78]. This finding may be explained by the fact that antifungal peptides can have an immediate effect on plant pathogens, while the CK group relies more on the antifungal substances produced by itself.

The correlation (Spearman correlation coefficient) between differential metabolism and antioxidant activity (POD, SOD, CAT) was conducted to shed more light on the potential relationship between antioxidant enzymes and differential metabolism shown in Figure 9 [72]. Up to 13 metabolic differentials were positively correlated with antioxidant enzyme capacity, while 16 metabolites presented negative correlations (|*r*|≥ 0.9 and *p* < 0.05). Among them, Try, Phe, ursolic acid, Pro, and gluconic acid were significantly positively correlated with POD enzyme activity, while arginosuccinate and senkyunolides were significantly positively correlated with SOD enzyme activity. Glycitein was significantly positively correlated with CAT enzyme activity. Phe, an amino acid necessary for plant growth and disease resistance, is synthesized via the oxalic acid pathway and subsequently incorporated into the Phe metabolic pathway via the Phe aminolytic enzyme (PAL) [79,80]. It was reported that treatment with a mixture of foliar fertilizers and pesticides significantly increased the antioxidant activity of POD enzymes and upregulated the shikimate-phenylpropane pathway [28]. The major products of the Trp metabolic pathway are indole compounds, and these compounds, as well as their derivatives, are ROS scavengers that work together with antioxidant enzymes such as SOD to remove excess ROS from plants [81]. Our findings on ursolic acid are consistent with a previous report that it could restore antioxidant enzyme POD activity in rice in response to salt stress [82]. Similarly, Pro is known as another potent ROS scavenger that plays a pivotal role in response to both biotic and abiotic stresses, and exogenous Pro was found to confer cold tolerance in *Chenopodium quinoa* seeds [83].

## 4. Conclusions

In this study, to explore the prevention and control mechanism of the antifungal peptide P852 (IC_50_, MFC) against Faba bean Fusarium wilt, the incidence and disease index of Fusarium wilt, growth indicators, antioxidant enzyme activity, and metabolomics determination were measured. (I) P852 exhibited an effective control effect on Faba bean Fusarium wilt under both field and laboratory conditions. (II) Compared to the CK group, IC_50_ and MFC of P852 could significantly enhance plant dry weight, fresh weight, plant height, and chlorophyll content, and reduce the oxidative damage in the Faba bean. Overall, the activities of antioxidant enzymes SOD, POD, CAT, chitinase, and β-1,3-GA were increased upon P852 treatment. (III) Different concentrations of P852 caused significant metabolic reprogramming in Faba bean leaves with Fusarium wilt. TCA cycle modifications, as well as modifications to a wide range of amino acids, organic acids, fatty acids, secondary metabolites, and biological pathways, were all brought about by P852 treatment, with an emphasis on bolstering antioxidant defense and enhancing the biosynthesis of isoquinoline alkaloids, betaine, and arginine. Generally, our research shows that the antifungal peptide P852 induces an immune and antifungal compound in the Faba bean plant, and that this peptide has a strong inhibitory effect on *F. oxysporum*. The findings with respect to how P852 changes the disease incidence, morbidity index, growth indicators, and metabolomic pathways provide valuable information that is highly relevant to crop yield and environmental safety. The mechanism underpinning the antifungal effects of P852 revealed in this study will shed new light on the production of green pesticides.

## Figures and Tables

**Figure 1 antioxidants-11-01767-f001:**
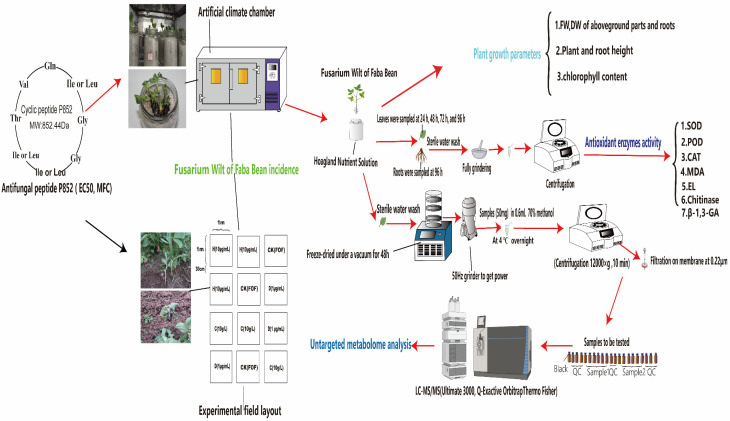
Flow chart for determining Fusarium wilt incidence assessment, plant growth indicators, antioxidant enzyme activity, and nontargeted metabolomics of Faba bean.

**Figure 2 antioxidants-11-01767-f002:**
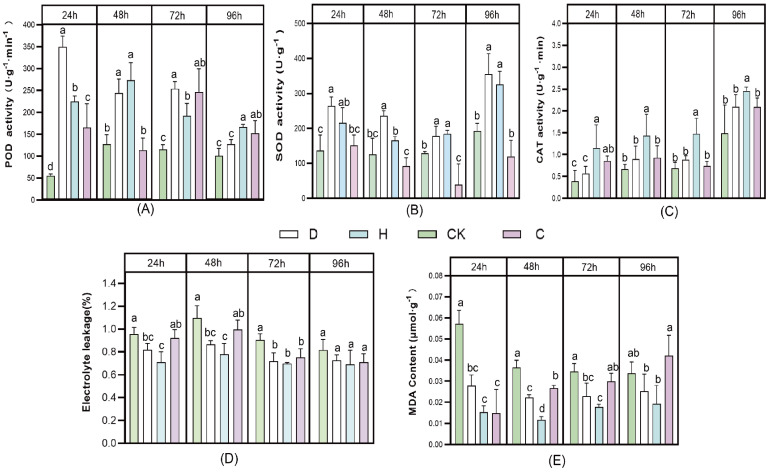
Effects of different treatments on the antioxidant capacity of Faba bean leaves with Fusarium wilt. (**A**) POD activity, (**B**) SOD activity, (**C**) CAT activity, (**D**) electrolyte leakage, and (**E**) MDA content. Values are means ± SDs (*n* = 3). Bars marked with different letters (a, b, c) represent significant differences (*p* < 0.05). The Fusarium wilt of Faba bean was inoculated with H (10 μg/mL), D (1 μg/mL), 10 g/L carbendazim (C), and seedlings treated with sterile water served as controls (CK).

**Figure 3 antioxidants-11-01767-f003:**
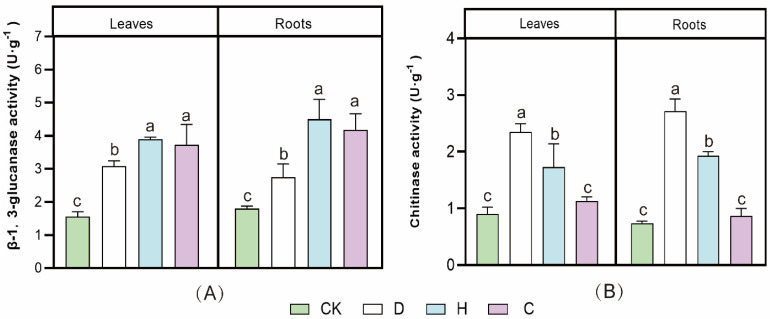
Effect of different concentrations of antifungal peptide P852 on Faba bean roots and leaves (**A**) β-1,3-glucanase and (**B**) chitinase activities. Values are means ± SDs (*n* = 3). Bars marked with different letters (a, b, c) represent significant differences (*p* < 0.05). The Fusarium wilt of Faba bean was inoculated with H (10 μg/mL), D (1 μg/mL), and 10 g/L carbendazim solution (C), and seedlings treated with sterile water served as controls (CK).

**Figure 4 antioxidants-11-01767-f004:**
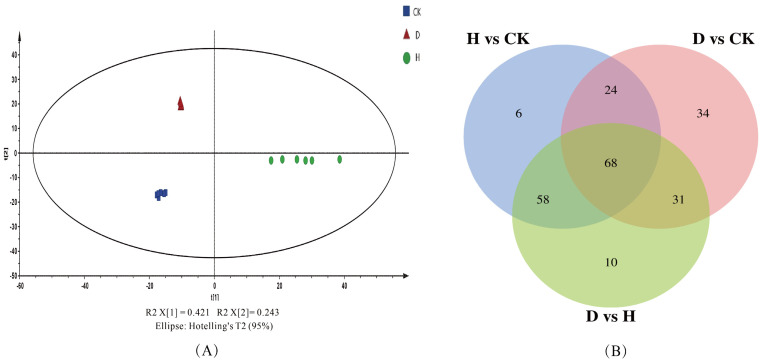
Principal component analysis (PCA, (**A**)) and Venn diagrams (**B**) of metabolic profiles in Faba bean leaves treated with antifungal peptide P852, H: P852 at 10 μg/mL, and D: 1 μg/mL of P852 and CK, with sterile water to treat Fusarium wilt of Faba bean, Venn diagrams with overlapping different color sections represent the same metabolites from different treatments.

**Figure 5 antioxidants-11-01767-f005:**
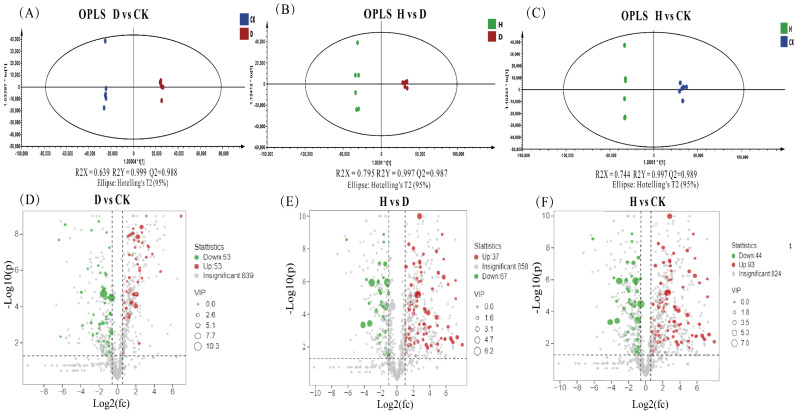
OPLS-DA score plots (**A**–**C**) and volcano plots (**D**–**F**) in response to different treatments: H: P852 of 10 μg/mL, D: 1 μg/mL of P852, and CK: sterile water to treat Fusarium wilt of Faba bean.

**Figure 6 antioxidants-11-01767-f006:**
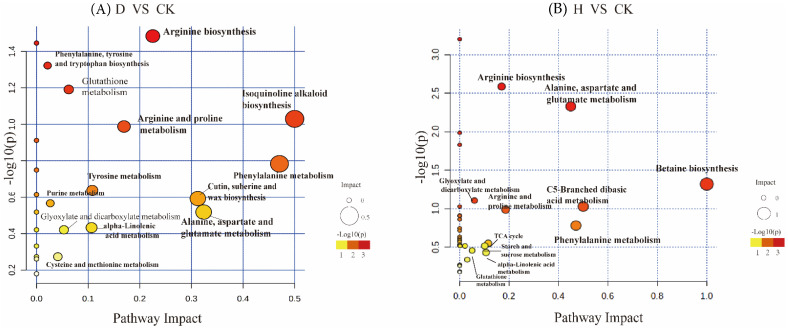
Pathway impact analysis containing (**A**) D vs. CK, (**B**) H vs. CK. The x-axis denotes the pathway impact; the y-axis represents pathway enrichment, the circle size denotes pathway to impact, and the circle color from red to yellow represents the *p*-value (*p* < 0.05) becoming smaller. H treatment: P852 at 10 μg/mL, D treatment: 1 μg/mL of P852, and CK: sterile water to treat Fusarium wilt in the roots of Faba bean.

**Figure 7 antioxidants-11-01767-f007:**
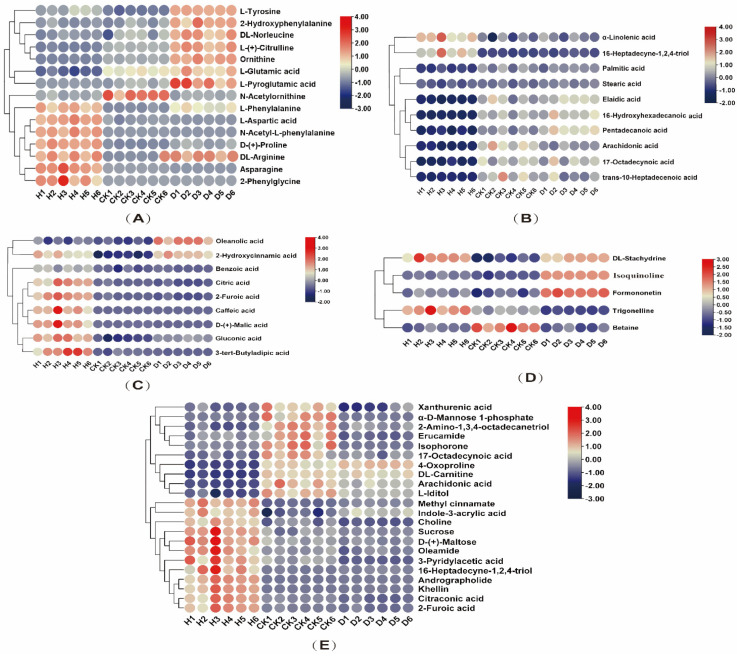
Heatmap analysis for the identified metabolites with a significant difference screened by OPLS-DA (VIP > 1) and ANOVA (*p* < 0.05). (**A**) Amino acid and its derivatives; (**B**) fatty acid; (**C**) organic acid; (**D**) secondary metabolites; (**E**) other. H treatment: P852 at 10 μg/mL, D treatment: 1 μg/mL of P852, and CK: sterile water to treat Fusarium wilt and the roots of Faba bean.

**Figure 8 antioxidants-11-01767-f008:**
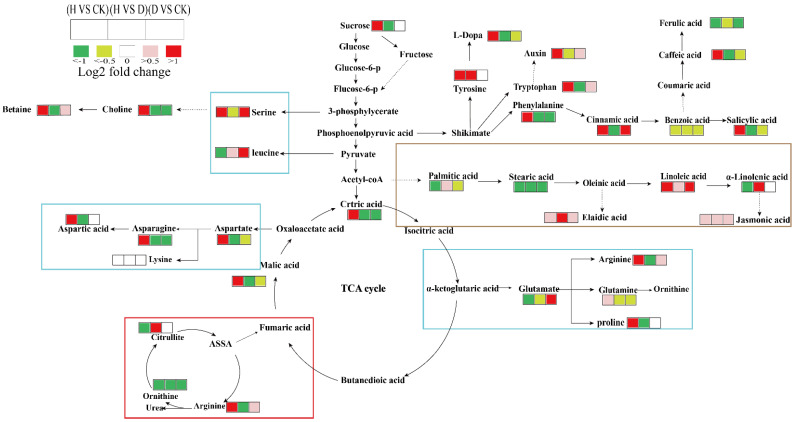
Schematic diagram of the proposed metabolic pathways of Fusarium wilt in Faba bean treated with H and D concentrations of P852. The fold change in the relative of primary metabolites in different groups was utilized to demonstrate the Fusarium wilt of Faba bean leaves; every chamber refers to D vs. CK, H vs. CK, H vs. D. The metabolites with red metabolites represent significant increases, and green chambers represent significant decreases (*p* < 0.05).

**Figure 9 antioxidants-11-01767-f009:**
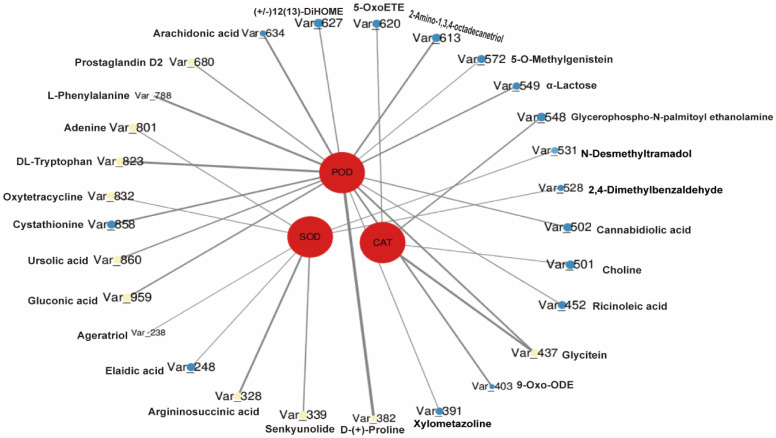
Network diagram between antioxidant capacity and metabolites. Red circles indicate different antioxidant enzymatic activities of guaiacol peroxidase (POD), superoxide dismutase (SOD), and catalase (CAT). Yellow represents a positive correlation between metabolites and antioxidant enzymes, and blue is a negative correlation; the size represents the *p*-value (the smaller the *p*-value, the larger the size), and the line connecting shows the circle denoting the correlation, with the thicker the line, the greater the correlation coefficient (|*r*| ≥ 0.9 and *p* < 0.05).

**Table 1 antioxidants-11-01767-t001:** Evaluation of Fusarium wilt in an artificial climate chamber and field test in Faba bean.

Different Treatments	Artificial Climate Chamber		Field Test	
	Incidence%	Disease Index	Incidence%	Disease Index
CK	91.67 ± 0.17 ^a^	18.33 ± 2.72 ^a^	83.33 ± 0.08 ^a^	5.9 ± 0.61 ^a^
D	58.33 ± 0.17 ^b^	9.58 ± 3.15 ^b^	52.78 ± 0.10 ^d^	2.9 ± 0.32 ^c^
H	58.33 ± 0.17 ^b^	6.67 ± 4.08 ^b^	55.56 ± 0.05 ^c^	2.5 ± 0.82 ^c^
C	66.67 ± 0.27 ^b^	10.42 ± 7.86 ^b^	58.40 ± 0.67 ^b^	3.5 ± 0.32 ^b^

Notes: Values within the same column with different letters are significantly different (*p* ≤ 0.05) by the LSD test, values are means ± SE (artificial climate chamber, plants replication *n* = 16; field test, 3 of fields were randomly selected for each same treatment group, and 12 plants were taken from each field, total plants *n* = 36), and the Fusarium wilt of Faba bean was evaluated in an artificial climate and field tests. The plants of Faba bean were inoculated with H (10 μg/mL) and D (1 μg/mL) of antifungal P852 and 10 g/L carbendazim solution (C), and seedlings treated with sterile water served as controls (CK).

**Table 2 antioxidants-11-01767-t002:** Chlorophyll content and physiological parameters in different treatment groups of Faba bean.

Different Treatments	Chlorophyll Content (mg/g)	Plant Height (cm)	Dry Mass of Stem with Leaves (g)	Dry Mass of Roots (g)	Fresh Mass of Stem with Leaves (g)	Fresh Mass of Roots (g)	Stem Diameter (cm)
CK	0.89 ± 0.45 ^c^	34.86 ± 1.60 ^c^	1.05 ± 0.13 ^b^	0.18 ± 0.01 ^c^	11.06 ± 0.84 ^b^	1.42 ± 0.89 ^b^	0.47 ± 0.06 ^b^
D	1.82 ± 0.08 ^ab^	46.17 ± 5.58 ^a^	2.70 ± 0.17 ^a^	0.69 ± 0.09 ^a^	18.30 ± 7.42 ^ab^	2.99 ± 1.24 ^ab^	0.70 ± 0.14 ^a^
H	2.43 ± 0.48 ^a^	45.43 ± 1.00 ^ab^	2.86 ± 0.46 ^a^	0.61 ± 0.61 ^a^	20.86 ± 7.81 ^a^	3.31 ± 0.1 ^a^	0.79 ± 0.05 ^a^
C	1.55 ± 0.11 ^bc^	36.87 ± 1.60 ^bc^	1.65 ± 0.49 ^b^	0.43 ± 0.06 ^b^	16.54 ± 0.83 ^ab^	2.68 ± 0.37 ^ab^	0.67 ± 0.10 ^a^

Notes: Value Notes: Values within the same column with different letters are significantly different (*p* ≤ 0.05) by the LSD test, values are means ± SE (artificial climate chamber, plants replication *n* = 16; field test, 3 of fields were randomly selected for each same treatment group, and 12 plants were taken from each field, total plants replication *n* = 36), Faba beans were evaluated in an artificial climate and field test, respectively. The plants of the Faba bean were inoculated at H (10 μg/mL) and D (1 μg/mL) concentration, 10 g/L carbendazim (C), and seedlings treated with sterile water served as controls (CK).

## Data Availability

The data presented in this study are available in the article and Appendix A.

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
