# Peer review of "Antifungal Peptide P852 Controls Fusarium Wilt in Faba Bean (Viciafaba L.) by Promoting Antioxidant Defense and Isoquinoline Alkaloid, Betaine, and Arginine Biosyntheses"

_antioxidants, 2022, doi:10.3390/antiox11091767_

Round 1

Reviewer 1 Report

This manuscript reports a study on the effects of the antifungal peptide P852 in controlling Fusarium wilt in faba bean.

Overall, this is a good work, performed using state of the art methods and equipment. The manuscript is well written and clearly presented. The conclusions are sound and well supported by the obtained results.

Given this, only minor corrections are needed:

L44: Replace “Also, faba bean is also a niche crop that can be used to detect and evaluate environmental pollution and toxicity [5].” with “Also, faba bean is a niche crop that can be used to detect and evaluate environmental pollution and toxicity [5].”.

Please explain in the introduction how faba bean can be used to detect and evaluate environmental pollution and toxicity.

Figures 1 and 5 are difficult to read, since they contains small size font. Please adjust.

L253: “GraphPad Prism 8.0 software (GraphPad Software Inc., San Diego, CA, USA).”. Something is missing from this sentence.

Author Response

Dear reviewer 1,

Thank you for your recognition of our work,kind comments and helpful suggestions. We have solved the problem you raised one by one, all our changes of manuscipt are displayed in red colour of MS Word. May happiness and health be with you always.

With kindest regards,

Yours Sincerely

Overall, this is a good work, performed using state of the art methods and equipment. The manuscript is well written and clearly presented. The conclusions are sound and well supported by the obtained results.

Given this, only minor corrections are needed:

Q: L44: Replace “Also, faba bean is also a niche crop that can be used to detect and evaluate environmental pollution and toxicity [5].” with “Also, faba bean is a niche crop that can be used to detect and evaluate environmental pollution and toxicity [5].”.

A: Thanks to the reviewer for the careful review, we have revised it in Line in 44.

Q: Please explain in the introduction how faba bean can be used to detect and evaluate environmental pollution and toxicity.

A: Dear reviewer, we have added this to Line in 44-47.

Also, Faba bean is a Faba bean is a model plant for environmental pollution that can be used to detect and evaluate environmental pollution and toxicity. The chromosome haplotype of Faba bean is 6 pairs of fairly large chromosomes, which are well suited for microscopic observation. 

Q: L253: “GraphPad Prism 8.0 software (GraphPad Software Inc., San Diego, CA, USA).”. Something is missing from this sentence.

 A: We have modified Line in 276-277.

GraphPad Prism software version 8.0 (GraphPad Software Inc., San Diego, California, USA).

Reviewer 2 Report

The paper is extensive, numerous experiments were carried out, which provided results interesting for science and agricultural praxis. Fusarium oxysporum is a worldwide pathogen of various plant species, among them Vicia faba. So this paper deals with an important problem. The results are presented in various interesting forms. The discussion is also very interesting. It is a pity that it is presented in one chapter of Results and Discussion - this form is more justified in short papers. For the above reasons, this manuscript should be published in MDPI, Antioxidants - this is a very suitable scope.

 However, the current version of the manuscript requires a large number of corrections and additions. Some of the errors are listed below. The chapter Material and methods is very poorly presented, there is a lack of a lot of data necessary for understanding how the experiments were performed and assessing their correctness. Much of this data was only presented in Results, and this explains some of the problems. However, this should be given in Methods (or at least where to find the information). The numerous mistakes indicate a certain carelessness. Perhaps, due to some carelessness in the notation, or because of large simplifications, certain passages are unclear. I believe that a correction is possible in a short time.

Remarks

Line 44 the text needs to be corrected

Line 62 it is F. oxysporum f. Sp. Fabae, it should be F. oxysporum f. sp. fabae

Line 64 Here you should specify the symptoms of the disease after infection of F. oxysporum in faba bean

Line 66   it is catalase (CAT) Chitinase, it should be catalase (CAT), chitinase ?

Line 73 in this case Fusarium shoud be in italic [fungal name italic, disease name Fusarium wilt -not italic], this rule should also be followed elsewhere in the text

Line 99  F. oxysporum was isolated from V. faba, - the year of isolation should be given. It is very important how long the cultures are stored, because after a long time since isolation, the fungi lose their many properties (physiological and biochemical and the degree of virulence).

Line 102 GenBank accession number CP010556.1 applies to Bacillus velezensis strain L-H15 and not P852, should be cited after Bacillus

Line 102 In a paper by Han et al. 2018 strain LH15 is Bacillus amyloliquefaciens L-H15, not Bacillus velezensis

Line 102-120 this chapter needs to be rewritten, the current form is unclear

Line 111 "Forage grass Microbe Laboratory" why grass not Grass

Line 116 How many plants were inoculated, this data is yet to be found in 'Results'

Line 116  1 × 106 should be specified because it is about the concentration of spores ??

Line 116  ‘Then the plant roots were excised and inoculated’ – what does excised mean in this case

Line 119 -120 it is not clear  1 μg / mL gives 50% maximal effect and 10 μg / mL is minimal concentration ???

Line 122  in the cited paper [9] states that 10 g / L of carbendazim was used, while the authors of this paper report 10 g / mL and cite [9]. This is a major difference and requires clarification.

In my opinion 10 g / mL it would be too much.

Line 126-128 the total annual accumulated temperature is 6883 C (by the way - how is this information helpful?), The annual average temperature is 17.8 C, - check if it is calculated correctly ?

Line 123 Field Trials - in this chapter only the terrain has been described, but there is no the issue of Field Trials is not mentioned at all. There is one sentence on this problem  in chapter 2.3 (line 136-137). This situation needs to be changed, corrected

Line 133-140 (chapter 2.3) requires a significant supplement, although it is a cited paper [1]. In its present form, it is not known at all which plant organs were taken for evaluation and which scale was used. There is no explanation for the formula given in line 140, it is difficult for the reader to look for explanations in the cited work [1]

Line 135 What does "Tissues were then taken from six faba bean plants to assess the disease" mean, is it the roots or the above-ground parts of plants with wilting symptome? The total number of plants tested is not given in Methods

Line 161 data are too general no data from how many plants the research material was collected and how much of this material was collected

Line 229 text correction required

Line 274  which means  '…. by 0.36, 0.36 times ’

Line 278, 281, 444   it is Fusarium Wilt, it should be Fusarium wilt

Line 279  the text needs to be corrected

Line 283- 284 the text needs to be corrected

Line 286 the text needs to be corrected

Line 288 it is Paeni Bacillus polymyxa,  it should be Paenibacillus polymyxa

Line 280, 333, 336, 346 g / mL carbendazim Line 298, 316 g / L carbendazim; which data is correct ?? [See note on line 122]

Line 297 the text needs to be corrected

Line 302 the text needs to be corrected

Line 306 and other places Fusarium should be written in italic

Line 315, 316 the text needs to be corrected

Line 328 the text needs to be corrected

Line 345    Faba bean in other places is faba bean - spelling should be standardized

Line 373 the text requires correction 'leaves of Faba faba bean plants'

Line 387 what is titin ???

Line 516 it is Polyscias Fruticosa it should be Polyscias fruticosa

Line 518     other organic acida or acids ??

Line 540 and 541 it should be stachydrine, and not stachydine ?

Line 581 instead of Quinoa should be rather quinoa, because it is not a botanical name of the genus but rather a popular name of  Chenopodium quinoa

Line 597 the text needs to be corrected

Line 676    Han, Y .; Deng, li ??

Line 682     Fang, zhongda ??

References - I see a significant problem in references. Well, in the Latin names for plants and fungi certain rules have to be followed. Genus should be written with a capital letter and the species with a small letter and should be written in italic. Editors should not require or allow any other spelling, especially typing the species with capital letters - and this is the case in numerous cited literature.

Author Response

Dear reviewer 2,

We appreciate your rigorous attitude, professional evaluation, and constructive suggestions. Thank you for spending a lot of valuable time patiently reviewing manuscripts. We have solved the problems you raised one by one, all our changes of manuscipt are displayed in red colour of MS Word. With best wishes for happiness in your life and work.

With kindest regards,

Dr. Yuzhu Han

Q: The results are presented in various interesting forms. The discussion is also very interesting. It is a pity that it is presented in one chapter of Results and Discussion - this form is more justified in short papers.

A: Thank you for your suggestion. However, we think that the results and discussion combined, can make the results and discussion analysis closely combined, continuity and logic may be better, and easy to read and author writing. This type of writing is permitted by the journal. Three of these references were found in this journal [1,2,3]; it was written using a combination of results and discussion.

We provided some similar manuscripts below with reference to a number of metabolomics and plant-related research papers in [1, 2 ,3 ,4, 5, 6,7],they also write the results in conjunction with the discussion.

Reference

  1. Tang, Y.-C.; Liu, Y.-J.; He, G.-R.; Cao, Y.-W.; Bi, M.-M.; Song, M.; Yang, P.-P.; Xu, L.-F.; Ming, J. Comprehensive Analysis of Secondary Metabolites in the Extracts from Different Lily Bulbs and Their Antioxidant Ability. Antioxidants 2021, 10, 1634, doi:10.3390/antiox10101634.
  2. StarzyÅ„ska-Janiszewska, A.; Fernández-Fernández, C.; Martín-García, B.; Verardo, V.; Gómez-Caravaca, A.M. Solid State Fermentation of Olive Leaves as a Promising Technology to Obtain Hydroxytyrosol and Elenolic Acid Derivatives Enriched Extracts. Antioxidants 2022, 11, 1693, doi:10.3390/antiox11091693.
  3. Saimaiti, A.; Huang, S.-Y.; Xiong, R.-G.; Wu, S.-X.; Zhou, D.-D.; Yang, Z.-J.; Luo, M.; Gan, R.-Y.; Li, H.-B. Antioxidant Capacities and Polyphenol Contents of Kombucha Beverages Based on Vine Tea and Sweet Tea. Antioxidants 2022, 11, 1655, doi:10.3390/antiox11091655.
  4. Pan, L.; Zhou, C.; Jing, J.; Zhuang, M.; Zhang, J.; Wang, K.; Zhang, H. Metabolomics Analysis of Cucumber Fruit in Response to Foliar Fertilizer and Pesticides Using UHPLC-Q-Orbitrap-HRMS. Food Chemistry 2022, 369, 130960, doi:10.1016/j.foodchem.2021.130960.
  5. Wang, Y.; Ren, W.; Li, Y.; Xu, Y.; Teng, Y.; Christie, P.; Luo, Y. Nontargeted Metabolomic Analysis to Unravel the Impact of Di (2-Ethylhexyl) Phthalate Stress on Root Exudates of Alfalfa (Medicago Sativa). Science of The Total Environment 2019, 646, 212–219, doi:10.1016/j.scitotenv.2018.07.247.
  6. Zhang, Y.; Huang, L.; Liu, L.; Cao, X.; Sun, C.; Lin, X. Metabolic Disturbance in Lettuce (Lactuca Sativa) Plants Triggered by Imidacloprid and Fenvalerate. Science of The Total Environment 2022, 802, 149764, doi:10.1016/j.scitotenv.2021.149764.
  7. Shen, S.; Huang, J.; Li, T.; Wei, Y.; Xu, S.; Wang, Y.; Ning, J. Untargeted and Targeted Metabolomics Reveals Potential Marker Compounds of an Tea during Storage. LWT 2022, 154, 112791, doi:10.1016/j.lwt.2021.112791.

Q: Line 44 the text needs to be corrected

A: We have modified and marked line 44 with a red color. 

Q: Line 62 it is F. oxysporum f. Sp. Fabae, it should be F. oxysporum f. sp. Fabae

A: Thank you for your careful discovery, we have modified it and marked it in red color in line 63.

Q: Line 64 Here you should specify the symptoms of the disease after infection of F. oxysporum in faba Bean

A:  Dear reviewer, we have added the symptoms of the symptoms of the disease after infection of F. oxysporum in faba Bean to the manuscript in line 66-69.

Symptoms of Faba bean Fusarium wilt caused by Fusarium oxysporum include yellowing and wilting of the leaves, which eventually turn black and die. The vascular bundle system of the root system and stems turns brown to black, with discoloration and decay at the base of the root system and stems.

Q: Line 66 it is catalase (CAT) Chitinase, it should be catalase (CAT), chitinase ?

A: Thank you for your reminder, we have changed it, in Line 71.

Q: Line 73 in this case Fusarium shoud be in italic [fungal name italic, disease name Fusarium wilt -not italic], this rule should also be followed elsewhere in the text

A: We have changed all the fungal names into italics in the manuscript.

Q: Line 99 F. oxysporum was isolated from V. faba, - the year of isolation should be given. It is very important how long the cultures are stored, because after a long time since isolation, the fungi lose their many properties (physiological and biochemical and the degree of virulence).

A: Fusarium oxysporum of faba bean was isolated on May 29, 2021. We conducted test in the lab in August 2021 and in the field in March 2022. Whether it is a lab experiment or a field test, we have all conducted experiments on the pathogenicity of Fusarium oxysporum on plants beforehand.

Q: Line 102 GenBank accession number CP010556.1 applies to Bacillus velezensis strain L-H15 and not P852, should be cited after Bacillus.

A: Thanks to the reviewers for their careful review, we have put the GenBank accession behind Bacillus in line 109.

Q: Line 102 In a paper by Han et al. 2018 strain LH15 is Bacillus amyloliquefaciens L-H15, not Bacillus velezensis

Q: Thank you very much for your careful review. We based on the recent discovery by scientists that Bacillus velezensis is a later heterotypic synonym of Bacillus amyloliquefaciens [8], fan et al. modified their strain Bacillus amyloliquefaciens FZB42 according to phylogenetic analysis, modified became Bacillus velezensis FZB42 [9]. Therefore, we adjusted the name of Bacillus in our manuscript from Bacillus amyloliquefaciens L-H15 to Bacillus velezensis L-H15.

Reference

  1. Dunlap, C.A.; Kim, S.-J.; Kwon, S.-W.; Rooney, A.P.Y. 2016 Bacillus Velezensis Is Not a Later Heterotypic Synonym of Bacillus Amyloliquefaciens; Bacillus Methylotrophicus, Bacillus Amyloliquefaciens Subsp. Plantarum and ‘Bacillus Oryzicola’ Are Later Heterotypic Synonyms of Bacillus Velezensis Based on Phylogenomics. International Journal of Systematic and Evolutionary Microbiology 66, 1212–1217, doi:10.1099/ijsem.0.000858
  2. Fan, B.; Blom, J.; Klenk, H.-P.; Borriss, R. Bacillus Amyloliquefaciens, Bacillus Velezensis, and Bacillus Siamensis Form an “Operational Group B. Amyloliquefaciens” within the B. Subtilis Species Complex. Frontiers in Microbiology 2017, 8.

Q: Line 102-120 this chapter needs to be rewritten, the current form is unclear

A: We have rewritten, please review Line in 108-129.

P852 antifungal peptide was extracted in the fermentation broth of Bacillus velezensis strain L-H15(GenBank accession number CP010556.1) [8]. The optimized P852 antifungal peptide’s conditions and extraction methods were as per the previously described [9]. The primary structure of P852 antifungal peptide was deduced to be glutamine (Gln)-isoleucine or leucine (Ile or Leu)-glycine (Gly)-glycine (Gly)-isoleucine or leucine (Ile or Leu)-isoleucine or leucine (Ile or Leu)-threonine (Thr)-valine (Val) sequence by thin-layer chromatography (TLC), matrix-assisted laser desorption time-of-flight mass spectrometry (MALDI-TOF-MS), electrospray-triple quadrupole mass spectrometry-time-of-flight tandem mass spectrometry (ESI-Q-TOF-MS), and nuclear magnetic resonance (NMR) [10].

Location of the experiment and inoculation method: The experiment was conducted in the Forage Microbe Laboratory, College of Animal Science and Technology, Southwest University. The plants of Faba bean were cultivated in an artificial climate chamber (Phase 1: at 25 ℃, 70% relative humidity (RH), 14,000 lx (light intensity); Phase 2: 75% RH, 0 lx, at 16 ℃) using Hoagland Nutrient Solution (pH = 6.5) with sterile quartz sands in a 500 mL tissue culture bottle until the plants grew to 7-8 true leaves. Then the plant roots were excised and inoculated with FOF at 1 × 10CFU/ mL spores suspension following the root-cut inoculation method [11], the P852 solution(sterile water) was poured into the root bases of Faba bean after two weeks.

Experimental design: The experiment consisted of one control (CK, treated with sterile water) and two treatments containing IC50 of P852 (1 μg/mL) and MFC (10 μg/mL, H) of P852. As described by Han [10], 10 g/L carbendazim was used as a positive control to treat Fusarium wilt tof in Faba bean leaves and roots [12].

Q: Line 111 "Forage grass Microbe Laboratory" why grass not Grass

A: We have revised, in Line 106 and 118.

Q: Line 116 How many plants were inoculated, this data is yet to be found in 'Results'

A: 36 faba bean strains were inoculated with Fusarium oxysporum, we have added to Line 123.

Q: Line 116 1 × 106 should be specified because it is about the concentration of spores ??

A: Thanks for the reminder, we have added specific information in line 123.

Q: Line 116 ‘Then the plant roots were excised and inoculated’ – what does excised mean in this case

A: Dear reviewer, excisions (means cut) the plant to make it easier for Fusarium oxysporum to infect Faba beans.

Q: Line 119 -120 it is not clear 1 μg / mL gives 50% maximal effect and 10 μg / mL is minimal concentration???

A: Dear reviewer, we have corrected that 1 μg/mL antifungal peptide P825 represents the IC50(The IC50 was defined as the concentration of antifungal agents that reduces the growth by 50% after 72 h incubation). 10 μg/mL antifungal peptide P852 refers to MFC (to determine minimum fungicidal concentration)[10].

Reference

10.Han, Y.; Zhao, J.; Zhang, B.; Shen, Q.; Shang, Q.; Li, P. Effect of a Novel Antifungal Peptide P852 on Cell Morphology and Membrane Permeability of Fusarium Oxysporum. Biochimica et Biophysica Acta (BBA) - Biomembranes 2019, 1861, 532–539, doi:10.1016/j.bbamem.2018.10.018.

Q: Line 122 in the cited paper [9] states that 10 g / L of carbendazim was used, while the authors of this paper report 10 g / mL and cite [9]. This is a major difference and requires clarification. In my opinion 10 g / mL it would be too much.

A: Sorry, due to negligence in work, we have now unified and corrected the concentration of carbendazim, 10 g/L of carbendazim is the concentration we applied.

Q: ine 126-128 the total annual accumulated temperature is 6883 C (by the way - how is this information helpful?)

A: We believe that providing the total annual accumulated temperature climate data can be used as a more detailed planting data reference for readers. Temperature and climate information is very important. It will affect the physiological and metabolic process of plants, and it also plays a significant role in Fusarium oxysporum-infected crops influences.

Q: The annual average temperature is 17.8 C, - check if it is calculated correctly?

A: We have checked and the data is correct.

Q: Line 123 Field Trials - in this chapter only the terrain has been described, but there is no the issue of Field Trials is not mentioned at all. There is one sentence on this problem in chapter 2.3 (line 136- 137). This situation needs to be changed, corrected

A: We have supplemented and added information on Field Trials on Line 139-143.

In this field, disease incidence occurred in Faba bean that were 90 days old, with the natural occurrence of Fusarium wilt. The experiment was conducted in a randomized block design, and repeated three times, with an area of 1 m2 in each plot (Figure 1). No insecticides, fungicides or herbicides were used during the growing period. Other management was based on local agronomic practices.

Q: Line 133-140 (chapter 2.3) requires a significant supplement, although it is a cited paper [1]. In its

present form, it is not known at all which plant organs were taken for evaluation and which scale

was used.

A: We have added relevant missing information in line 147-157.

The survey of the disease incidence and disease index of Faba bean’ s Fusarium wilt was conducted 16 plants were investigated in each treatment of the artificial climate chambers, 3 of fields were randomly selected for each same treatment group, and 12 plants were taken from each field using the five-point sampling method. The degree of disease incidence was divided into 5, levels: level 0-no symptoms; level 1- localized slightly diseased spots or slightly discolored spots on stem base or roots; level 2- diseased spots on stem base or main lateral roots, but not contiguous; level 3- diseased spots, discoloration or rot on 1/3~1/2 of stem base or roots; level 4- stem base surrounded by diseased spots or most of root system discolored and rotted; level 5- plants died. Incidence and disease index were calculated.

Q: There is no explanation for the formula given in line 140, it is difficult for the reader to look for explanations in the cited work [1]

A: Dear reviewers, we will add the formula explanations information in line 162-163.

Q: Line 135 What does "Tissues were then taken from six faba bean plants to assess the disease" mean, is it the roots or the above-ground parts of plants with wilting symptome? The total number of plants tested is not given in Methods

A: Dear reviewers, we have corrected and updated the method for investigating Fusarium wilt of faba beans in Line in 147-157.

Q: Line 161 data are too general no data from how many plants the research material was collected and how much of this material was collected

A: We have supplemented plant material information line in 190-235.

Q: Line 229 text correction required

 A: We have corrected on Line in 254.

Q: In line 237, correction required,

Line 274 which means '…. by 0.36, 0.36 times’

A: We have corrected in line 260, 297.

Q: Line 278, 281, 444 it is Fusarium Wilt, it should be Fusarium wilt

A: We have corrected at Line 297, 302, 305.

Q: Line 279 the text needs to be corrected

A: We have removed “and” Line in 302.

Q: Line 283- 284 the text needs to be corrected?

A: In 304 we have corrected.

Q: Line 286 the text needs to be corrected?             

A: We have corrected at Line in 310.

Q: Line 288 it is Paeni Bacillus polymyxa, it should be Paenibacillus polymyxa

A: We have changed it to Line in 312.

Q: Line 297, 333, 336, 346 g / mL carbendazim Line 298, 316 g / L carbendazim; which data is correct ??

A: We have revised and changed it on Line in 300,303,325,344,  364.

[See note on line 122]

Q: Line 297 the text needs to be corrected

A: We have changed at Line 323.

Q: Line 302 the text needs to be corrected?

A: We have corrected on line 328.

Q: Line 306 and other places Fusarium should be written in italic?

A: The Fusarium in the manuscript we have all changed to italics and has been marked in red

Q: Line 315, 316 the text needs to be corrected?

A: We have revised in 342-343.

Q: Line 328 the text needs to be corrected?

A: We have made changes at Line in 354.

Q: Line 345 Faba bean in other places is faba bean - spelling should be standardized ?

A: The Faba bean that we have in the manuscript has been unified.

Q: Line 373 the text requires correction 'leaves of Faba faba bean plants'

A: We have revised in Line in 401.

Q: Line 387 what is titin ???

A: We have revised in Line in 416-417.

Chitinase, the main component of the fungal cell wall, preventing mycelial growth, causing rough deformities, and even complete cell lysis, thereby mitigating fungal infestation [44].

Q: Line 516 it is Polyscias Fruticosa it should be Polyscias fruticosa

A: We have corrected in Line 544.

Q: Line 518 other organic acida or acids ??

A: Due to our carelessness, it has now been corrected on Line in 547

Q: Line 540 and 541 it should be stachydrine, and not stachydine?

A: Line in 566-570 we have revised.

Q; Line 581 instead of Quinoa should be rather quinoa, because it is not a botanical name of the genus

but rather a popular name of Chenopodium quinoa .

A: Dear reviewer, in line in 609 we have revised.

Q: Line 597 the text needs to be corrected

A: Line in 607 we have corrected.

Q: Line 676 Han, Y .; Deng, li ?? , Line 682 Fang, zhongda ??

A: Line in 706, 712, we have corrected.

Q: References - I see a significant problem in references. Well, in the Latin names for plants and fungi

certain rules have to be followed. Genus should be written with a capital letter and the species with a small letter and should be written in italic. Editors should not require or allow any other spelling, especially typing the species with capital letters - and this is the case in numerous cited literature.

A: we have corrected the formatting of the reference in italics.

Reviewer 3 Report

The submitted manuscript presents the results of antifungal effects of a peptide P852 on growth of Fusarium sp. and development of Fusarium wilt. Manuscript has very good construction and all parts are presented quite well. The quality of ilustrations and graphs is very good. The data are  interesting and provide an excellent starting point for a future studies on antifungal effect of  this peptide as a biocontrol agent. This idea should be presented in the conclusions.

Minor points:

1) Fusarium name should be written in italic (all manuscript), also Fusarium wilt.

2) L24: chitinase

3) L39: moderate climate

4) L66: ..., chitinase

5) L75: are antimicrobial

6) L77: Bacillus spp.

7) L88-89: conducted and noted 

Author Response

Dear reviewer 3,

 We are grateful for your valuable suggestions and recognition. Thank you for acknowledging the significance of our research on the use of antifungal peptides for the control of Fusarium oxysporum. At the same time, we have received a lot of help from you. Amendments were highlighted with RED in the revised manuscript.

Yours Sincerely

Dr. Yuzhu Han

Q:1) Fusarium name should be written in italic (all manuscript), also Fusarium wilt.

Thank you, for pointing out that we have modified the problem of Fusarium to be italicized. But we think Fusarium wilt is the correct format. We looked up the relevant literatures again, they are also Fusarium wilt written like this way [1,2].

Reference

[1] Jiaxing Lv, Yan Dong, Kun Dong, Qian Zhao, Zhixian Yang, Ling Chen. Intercropping with wheat suppressed Fusarium wilt in faba bean and modulated the composition of root exudates. Plant and Soil, 2020. https://doi.org/10.1007/s11104-019-04413-2.

[2] Zhao Q, Chen L,Dong K,Dong Y,Xiao J X. Cinnamic Acid Inhibited Growth of Faba Bean and Promoted the Incidence of Fusarium Wilt. Plants, 2018, 7, 84. https://doi.10.3390 /plants 7040084. 

Q: 2) L24: chitinase

A: Dear reviewer, we have revised it in Line in 24 and marked it in red.

Q:3) L39: moderate climate

A: Thanks to you for being careful to review, we have revised it in Line in 39.

Q:4) L66: ..., chitinase

A: We have modified in Line 71.

Q:5) L75: are antimicrobial

A: we have revised it in Line 80-81.

Q: 6) L77: Bacillus spp.

A: we have revised in Line 82.

Q:7) L88-89: conducted and noted 

A: we have revised it in Line in 94-95.
